# Beneficial Effects of Spirulina Consumption on Brain Health

**DOI:** 10.3390/nu14030676

**Published:** 2022-02-05

**Authors:** Teresa Trotta, Chiara Porro, Antonia Cianciulli, Maria Antonietta Panaro

**Affiliations:** 1Department of Clinical and Experimental Medicine, University of Foggia, 71121 Foggia, Italy; teresa.trotta@unifg.it (T.T.); chiara.porro@unifg.it (C.P.); 2Department of Biosciences, Biotechnologies and Biopharmaceutics, University of Bari, 70125 Bari, Italy; antonia.cianciulli@uniba.it

**Keywords:** spirulina, phycocyanin, astrocytes, microglia, Parkinson’s disease, Alzheimer’s disease, multiple sclerosis, neuroinflammation, neurodegeneration diseases

## Abstract

Spirulina is a microscopic, filamentous cyanobacterium that grows in alkaline water bodies. It is extensively utilized as a nutraceutical food supplement all over the world due to its high levels of functional compounds, such as phycocyanins, phenols and polysaccharides, with anti-inflammatory, antioxidant, immunomodulating properties both in vivo and in vitro. Several scientific publications have suggested its positive effects in various pathologies such as cardiovascular diseases, hypercholesterolemia, hyperglycemia, obesity, hypertension, tumors and inflammatory diseases. Lately, different studies have demonstrated the neuroprotective role of Spirulina on the development of the neural system, senility and a number of pathological conditions, including neurological and neurodegenerative diseases. This review focuses on the role of Spirulina in the brain, highlighting how it exerts its beneficial anti-inflammatory and antioxidant effects, acting on glial cell activation, and in the prevention and/or progression of neurodegenerative diseases, in particular Parkinson’s disease, Alzheimer’s disease and Multiple Sclerosis; due to these properties, Spirulina could be considered a potential natural drug.

## 1. Introduction

Spirulina is an undifferentiated microscopic, filamentous, spiral-shaped cyanobacterium (blue–green alga) [1] that is capable of growing naturally in alkaline and saline environments and double its biomass every 2–5 days. Spirulina is the name utilized to describe mainly two species of cyanobacteria: *Spirulina platensis*, and *Spirulina maxima* [2]. For centuries, Aztec and Maya civilizations used Spirulina as a primary food source [3]; nowadays it is considered safe for human consumption, having very low toxicity [4], and represents more than 30% of the world’s production of microalgae biomass. It is cultivated worldwide as a fundamental ingredient in many nutraceutical formulations or as food [5]. In 1996, the World Health Organization declared Spirulina the best food for the future thanks to scientific research that reported its high content of proteins and natural vitamins [3]. More recently, the United States Food and Drug Administration (FDA) awarded Spirulina the “Generally Recognized as Safe (GRAS)” status. Moreover, the Dietary Supplements Information Expert Committee (DSI-EC) of the United States Pharmacopeial Convention (USP) granted ‘Class A’ to Spirulina after rigorous analysis of clinical case reports, animal toxicological data and adverse event reports, thus making it safe for human consumption [6] when grown under controlled conditions [4,7].

The protein content of Spirulina varies between 60 and 70% of its dry weight [8], and the most important proteins for food applications are phycobiliproteins, namely phycocyanin and allophycocyanin, which have the same chromophore group, and phycoerythrin. The phytochemical mostly represented is phycocyanin, a deep blue colored protein due to the open tetrapyrrole chromophore, phycocyanobilin (PCB), covalently joined to the apoprotein [9] and able to harvest light energy that the organism uses to drive ATP production [10]. In addition, Spirulina contains polysaccharides, polyunsaturated fatty acids, vitamins (especially B vitamins) [11], carotenoids [12] as well as several minerals such as Na, K, Ca, Fe, Mn, Se, Mg and Zn [8].

There is scientific evidence attesting to its hypolipemic, antihypertensive, antidiabetic, neuroprotective, antianemic, anticarcinogenic, hepatoprotective as well as anti-bacterial, anti-viral and immunomodulatory properties [9,13,14,15,16]. It has been demonstrated that Spirulina is able to enhance phagocytosis in macrophages [17,18,19] as well as induce cellular and humoral adaptive immunity [17,20,21]. Spirulina regulates key cytokines, including IL-1β, IL-2, IL-4, IL-6, IL-10, TNF-α and IFN-γ [9]. The anti-inflammatory properties of Spirulina may be due to the inhibition of cyclooxygenase-2 (COX-2) activity [22], but the protein extract of Spirulina also shows chelating ability, reduces lipid peroxidation and DNA damage, and scavenges free radicals [23,24]. The antioxidant and free radical-scavenging properties of Spirulina may be attributed to its phycocyanin content, which in turn ascribes its antioxidant activity to PCB [25].

C-Phycocyanin (C-PC) is formed by the α and β polypeptide subunits and PCB (4.7% of the molar mass of C-PC), a linear tetrapyrrolic chromophore covalently bound to C-PC [26] that induces an essential light harvesting in blue–green algae such as *Spirulina platensis* [27]. When administrated in vivo by oral or parenteral routes, C-PC is degraded by proteolysis in PCB or PCB-linked peptides that probably are responsible for the pharmacological actions reported for this biliprotein [28,29]. It was postulated that PCB would inhibit NADPH oxidase since the chemical structure of PCB is similar to that of bilirubin, which is a highly specific inhibitor of NADPH oxidase [30,31]. In addition to phycocyanin, allophycocyanin has strong antioxidant activity, proving to be higher than phycocyanin as scavenging peroxyl radicals whereas phycocyanin is better at scavenging hydroxyl radicals [32].

Two novel anti-inflammatory peptides with amino acid sequences of LDAVNR (P1, 686 Da) and MMLDF (P2, 655 Da) were purified from the enzymatic hydrolysate of *Spirulina maxima*. Both P1 and P2 exhibited a suppressive effect on IL-8 expression in histamine-stimulated EA.hy926 endothelial cells, and on the production of intracellular reactive oxygen species (ROS) in the mast and endothelial cells [33].

Spirulina also contains carotenoids, which are a group of precious natural pigments. The carotenoids in Spirulina are as high as 4000 mg/kg, among which β-carotene predominates [34], and is considered an efficient membrane antioxidant showing antioxidant properties and providing protection against singlet oxygen-mediated lipid peroxidation damage [15]. In addition, Bai et al. showed that β-carotene directly blocked the intracellular accumulation of ROS in murine macrophage cell line RAW264.7 [35] and inhibited the expression of the inflammatory genes in lipopolysaccharide (LPS) stimulated RAW264.7 cells and LPS-treated mice [36].

Although different studies have tried to elucidate the signaling pathways involved in the antioxidant and anti-inflammatory effects of Spirulina, the molecular mechanism(s) by which Spirulina performs these actions is still unknown. It seems that it effectively controls the ERK1/2, JNK, p38 and IκB pathways. In Ogg1-KO mice, after UVB exposure, *S. platensis* suppressed the phosphorylation of p38 MAPK, SAPK/JNK and ERK, indicating that it possesses various effective sites for the inhibition of skin tumor development [37]. Kim et al. showed that phycocyanin-induced expression of heme oxygenase-1 (HO-1) is moderated by the PKC α/β II-Nrf-2/HO-1 pathway, and inhibits UVB-induced apoptotic cell death in primary skin cells [38]. Another study by Khan et al. demonstrated that C-PC ameliorated the recovery of cardiac function during ischemia-reperfusion (I/R)-induced myocardial injury by increasing the activation of ERK1/2 and the expression of Bcl-2 and by reducing the activation of p38 MAPK and caspase-3 [39].

## 2. Role of Spirulina in the Brain

In the last few years, a large number of studies have demonstrated that Spirulina not only has beneficial effects on the development of the neural system but also neuroprotective effects through attenuating oxidative stress and antioxidant properties [40,41,42,43]. It is a rich source of PCB, an inhibitor of NADPH oxidase, which is implied to be a contributor to oxidative stress in a variety of neurological/neurodegenerative diseases [38].

During aging and neurodegeneration, there is a decrease in normal antioxidant and anti-inflammatory defense mechanisms, making the brain more susceptible to the detrimental effects of oxidative stress [44,45,46]. Indeed, most neurological disorders such as Alzheimer’s disease (AD), Parkinson’s disease (PD), inflammatory injuries and senility are the result of age-linked inflammation and/or oxidative stress increase [44,45].

It was demonstrated that treatment with Spirulina-enriched diets increases cerebellar glutathione (GSH) levels, reduces malondialdehyde (MDA) levels, reduces pro-inflammatory cytokines, and ameliorates both spatial and motor learning in aged rats [24,46,47]. In cerebral ischemia, a condition marked by cerebral hypoxia with the generation of free radicals, ROS or reactive nitrogen species (RNS) and energy crisis, Spirulina treatment evidenced neuroprotective effects with progressive decline in TUNEL positive cells and caspase-3 activity in the ischemic hemisphere [48]. In Sprague Dawley rats with partial crush injury induced at the level of T12, supplementation of Spirulina revealed an advance in the fine ultrastructure of spinal cord gray matter when compared to the control group, thereby indicating the neuroprotective potential of Spirulina in mitigating the effects of spinal cord injury and causing functional recovery [49].

Among the agents responsible for the production of oxidative stress and decline in neuronal functions, iron is one of the most important. Increased accumulation of iron in specific brain areas seems to be implicated in the neuropathology of various neurodegenerative disorders such as PD and AD [50,51,52,53]. Although the etiology of neurodegenerative diseases is not yet well established, accumulating evidence has demonstrated that iron-dependent oxidative stress induces depletion of antioxidants in the brain [54,55]. Bermejo-Bescós et al. studied, in a neuroblastoma cell line (SH-SY5Y), the action of the *S. platensis* protein extract on oxidative stress caused by iron, characterized by the exhaustion of the enzyme activities of the GSH pathway, reduction in GSH and increase in the glutathione disulfide (GSSG)/GSH ratio. GSH is an antioxidant enzyme which, by the action of the enzyme glutathione peroxidase (GPx), reduces H_2_O_2_ to H_2_O and oxidizes to GSSG, which in turn is reduced to GSH by the enzyme glutathione reductase (GR) [56]. Fe^2+^ can lead to the production of hydroxyl or alkoxy radicals, which both the *S. platensis* protein extract and phycocyanin have been shown to be able to eliminate, preserving the activity of antioxidant enzymes. Furthermore, Spirulina has been shown to be an iron chelator, preventing reaction with oxygen or peroxides. This study showed that *S. platensis* protein extract protects the activity of total GPx, GPx-Se and GR, and increases reduced GSH thanks to the scavenger activity of free radicals and iron chelator ability, thus increasing cell viability [57].

Nutrition is one of the most important environmental factors that influence brain development [58,59]; several studies showed that maternal malnutrition affects brain development and probably represents a leading factor in the pathogenesis of neurological and psychiatric disorders in an offspring [60,61,62]. A more recent article demonstrated that Spirulina supplementation to protein-deprived mothers intensifies reflex maturation, reduces mortality, and ameliorates behavioral and cognitive problems in the offspring; moreover, it attenuated protein malnutrition-induced microgliosis, which may provide for the restoration of neurocognitive outcomes [63]. In another study, Sinha et al. observed that maternal dietary supplementation with Spirulina during pregnancy, and lactation to protein-malnourished (PMN) rats reduced astrocytic and microglial activation in F1 progeny. In particular, Spirulina prevented PMN-induced astrogliosis by effectively reducing the astrocytic population labeled with glial fibrillary acidic protein (GFAP) and S100β, and with enlarged cell body and thick processes, a morphology that characterizes activated astrocytes. In addition, it was reported that Spirulina caused a reduction of CR3 and MHC II expression on microglia cells that lose the typical shape of activated phagocytic cells. Finally, Spirulina treatment attenuated MDA levels, modulated superoxide dismutase (SOD) and catalase (CAT) activities, increased cerebral cortical thickness, improved neuronal morphology, and increased dendritic arbor complexity, thus enhancing behavioral and cognitive functions [64].

Pavón-Fuentes et al. evaluated if PCB protected PC12 neuronal cells against oxygen glucose deprivation (OGD) plus reperfusion, and its protective role in a rat model of endothelin-1-induced focal brain ischemia. PCB treatment significantly reduced brain infarct volume, limited exploratory behavior impairment, and preserved viable cortical neurons in ischemic rats in a dose-dependent manner with respect to the vehicle group. Furthermore, PCB at high doses restored the expression levels of myelin basic protein and the enzyme CNPase in ischemic rats [65].

In a recent article by Song et al., the selenium-containing protein from selenium-enriched *S. platensis* (Se-SP) demonstrated a high ability to inhibit OGD-induced neurotoxicity and apoptosis of primary hippocampal neurons in vitro. Se-SP was showed to suppress OGD-induced DNA damage by arresting ROS accumulation in neurons and improving mitochondrial dysfunction via balancing Bcl-2 family expression. In addition, inhibition of the mitochondrial permeability transition pore (mPTP) by CsA (an mPTP inhibitor) with consequent decrease of OGD-induced ROS generation, oxidative damage and mitochondrial membrane potential loss was observed [66].

Another study showed that 70% ethanol extract of *S. maxima* (SM70EE) prevents trimethyltin (TMT) neurotoxicity, such as oxidative stress, neurotransmitter dysfunction and neuronal death. TMT induces the release of the brain-derived neurotrophic factor (BDNF), which phosphorylates tropomyosin-related kinase receptor type B (TrkB), resulting in the expression of the cyclic AMP responsive element binding protein (CREB) and increase in BDNF, an important factor for neuronal survival. Furthermore, TMT increases acetylcholinesterase (AChE), which in turn reduces the expression of BDNF. The neurotoxicity that is triggered stimulates PARP cleavage, which is a protein associated with apoptosis, and therefore causes the death of neuronal cells. SM70EE is able to inhibit AChE activity and PARP cleavage, and increase the expression of p-CREB, p-TrkB and BDNF, promoting the activation of BDNF/CREB neuroprotective signaling pathways in HT-22 neuronal cells [67].

Moreover, in PC12 cells, SM70EE, C-PC and chlorophyll-a reduced amyloid-beta (Aβ)1-42-induced neuronal cell death and PARP cleavage, attenuated oxidative stress by enhancing anti-oxidant enzyme level, increased the level of BDNF, which has a critical role in neuronal survival and neuroprotection, and inhibited amyloid precursor protein (APP) processing [68].

### Effects of Spirulina on Glial Cells Activation

Neuroinflammation is an effective endogenous defense that protects the central nervous system (CNS) and aims not only to eliminate threats but also restore homeostasis [69], although it is widely proven that extended neuroinflammatory events can lead to the progressive neuronal damage described in many neurodegenerative diseases [70]. An important role for chronic neuroinflammation is fulfilled by neuroimmune cells, astrocytes and microglia, which play a double role, both pro- and anti-inflammatory, and are involved in different functions under physiological and pathological conditions [71].

Astrocytes represent structural and nutritional support for neurons, are involved in the development of the neural circuit, have an important role to play in the regulation of fluid and ion homeostasis and neurotransmitters, in the control of CNS microenvironment, and in controlling the blood–brain barrier (BBB) [72].

Microglia are responsible for the first line of immune defense in the CNS [73]. In the adult CNS, microglia are associated with an immunosurveillance state, and maintenance of homeostasis. Microglia are important for the removal of dead cell debris and abnormally accumulated proteins [74]. They are fundamental to the survival and proliferation of neurons [75,76], and in addition, microglia monitor synaptic elements and networks, responding to dyshomeostasis by producing or eliminating synaptic elements and modulating neuronal activity [72]. Usually, communication between microglia and astrocytes is important to maintain the integrity of the neural circuit. They respond to neuronal damage with an intricate process that involves proliferation, morphological alterations, production of mediators, and engulfment of cells and subcellular elements [72]. Notably, activated microglia is recognized to cause the activation of astrocytes, which in turn can also liberate signaling molecules that act on microglia [77].

Altered glial cell activity is a major source of free radicals under altered neurological conditions [78], and a vicious circle mechanism is thus created, in which neuronal death or dysfunction induces the activation of glial cells, which in turn further traumatizes neurons. Persistent activation of microglia and astrocytes elevates inflammatory mediators like chemokines and cytokines [79], the ROS and RNS release, with neurotoxic effects and responsible for chronic neurological/neurodegenerative disorders [80,81,82].

Several studies have demonstrated the protective effects of Spirulina or its components, in particular C-PC, against glial activation (Figure 1). In male Wistar rats treated with tributyltin chloride (TBTC), an environmental pollutant and potent biocide, C-PC effectively reduced ROS generation, decreased astroglia activation, counteracting the morphological alteration, and upregulation of GFAP, indices of astroglia activation. In addition, it significantly restored microglial activity, adjusting the expression of cellular marker CD11b and the formation of LC3-II and beclin (markers of autophagy), reduced by treatment with TBTC, apart from its considerable protective effect on redox signaling, stress proteins and inflammatory molecules [83].

The protective role of C-PC on the permeability of the BBB was evaluated in terms of BBB interruption induced by TBTC. C-PC failed to fully restore BBB disruption, although it was able to restore p-glycoprotein (PGP) and claudin-5 levels and, partially, enzyme activity of tissue remodeling such as calpain and matrix metalloproteinase [83]. The protective role of C-PC on damaged BBB was also suggested in a small vessel disease (SVD) study, in which McCarty reported that oxidative stress generated by NADPH oxidase may contribute to BBB damage, and PCB from Spirulina may improve the course of SVD by suppressing the activity of NADPH oxidase [84].

Moreover, oral administration of C-PC also decreased the activation of microglia and astrocytes caused by kainic acid in rats [85], evaluated by studying, in the hippocampus, the expression of the heat shock protein 27kD on astrocytes and the peripheral benzodiazepine receptor on microglia. The authors suggested that PCB, when released from C-PC in vivo, can be absorbed, cross the BBB, and by this way, exert antioxidant effects in the hippocampus, suppressing ROS production from activated microglia [85,86,87]. The possibility that the active components of Spirulina are able to cross the BBB and perform their function in the brain is suggested by in-vivo experimental evidence (Table 1). Only Mitra et al. evaluated the bioavailability of C-PC in homogenates of cerebral cortical tissue by measuring absorbance at 620 nm in Wistar rats that received intraperitoneal injections of C-PC [83].

Min et al. observed that C-PC intranasal administration to rats with ischemic brain reduced infarct size and improved behavioral deficits. C-PC also attenuated apoptosis and ROS of oxidized astrocytes. In the 3D oxidized astrocyte model, C-PC upregulated antioxidant enzymes such as SOD and CAT, and downregulated inflammatory factors IL-6 and IL-1β and glial scar. Moreover, C-PC induced the expression of nerve growth factor (NGF) and BDNF [98], two important neurotrophic factors described as neuroprotective and neurotrophic under neurodegenerative environments [99,100]. It is known that BDNF exerts positive effects such as anti-apoptosis, anti-inflammation, anti-neurotoxicity and neural regeneration. It was seen that astrocyte-derived BDNF supports OPC maturation under hypoxic conditions in vitro and oligodendrogenesis after white matter damage in vivo, also indicating a positive role of C-PC in myelin regeneration [101]. Another study showed that Spirulina has a significant protective effect on hippocampus neural progenitor cells (NPCs) against LPS-induced acute systemic inflammatory response. LPS insult causes increased astrogliosis with prominent activation of GFAP in existing cells and decreased proliferation of NPCs. However, diet supplemented with Spirulina before LPS administration prevents LPS-induced reduction in NPC proliferation and astrogliosis, evaluated by studying the expression of GFAP [102].

## 3. Beneficial Effects of Spirulina in Neurodegenerative Diseases

Neurodegenerative diseases are characterized by a series of aberrant protein folding events, which cause the formation of amorphous aggregates or amyloid fibrils [103]. There is considerably strong evidence to clarify that inflammation and oxidative stress play a fundamental role in the onset and progression of neurodegenerative diseases [45], and adequate scientific evidence has shown the beneficial effects of Spirulina in PD, AD and Multiple Sclerosis (MS).

### 3.1. Spirulina in PD

PD is a chronic, progressive and neurodegenerative disorder, the etiology of which depends on many factors. It affects mainly older people; the onset of PD increases with age, being 1–2% after 60–65 years [104]. The pathophysiology underlying most PD patients is complex, and currently, only partially understood. Age is considered to be the predominant risk factor for PD [105], but exposure to pesticides and brain trauma also increase the possibility of disease onset [106]. In addition to environmental factors, genetic factors are fundamental in PD; in fact, at least 23 loci and 19 genes that cause diseases and several genetic risk factors have been recognized [107]. Variants of genes such as SNCA (encoding α-Synuclein (α-Syn), including the A53T mutation), GBA (glucosylceramidase β, encoding GBA protein), LRRK2 (leucine-rich repeated kinase 2, encoding LRRK2 protein) and MAPT (encoding microtubule associated protein Tau) have been found to raise the possibility of PD onset [108].

The clinical features of PD are manifested by motor symptoms such as bradykinesia, rigidity, tremors and postural instability [109], and non-motor symptoms such as depression, constipation, sleep disturbances and dementia that can appear earlier and significantly affect the quality of life [110]. This disease presents a massive degeneration of dopaminergic neurons in the substantia nigra pars compacta and appearance of Lewy bodies (LB) in the remaining neuronal cells [111]. LB are primarily composed of aggregated forms of the presynaptic protein α-Syn [111,112], and the aberrant aggregation of α-Syn is believed to participate in PD pathogenesis [113]. Nevertheless, the molecular mechanisms leading to α-Syn aggregation are still not known.

It has been widely accepted that brain inflammation plays a key role in neurodegenerative disorders such as PD. Post-mortem clinical studies demonstrated high levels of pro-inflammatory molecules in the brains of individuals with PD, as well as in the cerebrospinal fluid [114,115]. Moreover, α-Syn has been shown to increase the expression of COX-2 [116] and the activity of microglial NADPH oxidase via CD11b [117]. Interestingly, C-PC proved to be an effective inhibitor of A53Tα-Syn and Aβ40/42 fibril formation. C-PC and α-Syn interact through unstable interactions, indicating that perhaps transient interactions can hinder fibril formation [118].

Diets rich in foods with anti-inflammatory and antioxidant effects can modulate this neuroinflammation, and Spirulina has been shown to have a neuroprotective role (Figure 2). In vitro and in vivo studies have shown that the antioxidant components of Spirulina, such as phycocyanin, can block or slow down oxidative damage by decreasing the accumulation of ROS [119], activating the antioxidant enzyme systems of CAT, SOD and GPx [120]. Studies on experimental PD models argue that dopamine neurons are very susceptible to both oxidative stress and inflammation [121,122], and activation of microglial cells is thought to be involved in the pathogenesis of neurodegenerative diseases [88]. Recent evidence has shown that neuronal cells constitutively express CX3CL1, which binds to a G protein-coupled receptor CX3CR1 present in microglia, controlling microglia activation and lowering levels of IL-1β, TNF-α and IL-6 [123,124]. An interesting aspect is that oral treatment with C-PC shows its effect in the hippocampus, crossing BBB [85]. In rats fed a diet enriched with Spirulina prior to a 6-hydroxydopamine (6-OHDA) lesion, an improvement in the recovery of striatal dopamine-positive TH nerve fibers and SNpc-positive TH neurons was found one month after injury. In addition, a decrease in the number of activated microglia (M1 phenotype, pro-inflammatory), as determined by expression of the major histocompatibility complex II (MHC II), a biomarker related to the activation of M1 microglia, was observed [125]. More recently, Pabon et al. studied the effects of Spirulina on the inflammatory response in a α-Syn rat model of PD [126]. The authors showed that Spirulina was neuroprotective in this α-Syn model of PD, observing a major number of TH+ and NeuN+ cells and a decrease in the number of activated microglial cells as evaluated by MHC II expression. The reduction in microglia activation may be caused, in part, by Spirulina’s capacity to enhance expression on the microglia of the fractalkin receptor (CX3CR1), which is neuroprotective in a 6-OHDA model of PD [126].

*S. maxima* showed a protective effect in response to 6-OHDA-induced toxicity in the rat striatum, inducing a reduction in NO, ROS and lipoperoxidation levels in the striatum [89], and pretreatment with Spirulina did not completely prevent the DA-consumption effect of 1-methyl-4-phenyl-1,2,3,6-tetrahydropyridine (MPTP) and arrest oxidative stress [90]. Pretreatment with polysaccharide derived from *S. platensis* significantly increased immunoreactive staining and mRNA expression of the dopamine transporter (DAT) and tyrosine hydroxylase, the rate-limiting enzyme in dopamine synthesis, in the substantia nigra of MPTP-treated mice. Moreover, in contrast to the activity of monoamine oxidase B (MAO B), the activities of SOD and GPx in the serum and midbrain were increased significantly, suggesting that the antioxidant capabilities of this polysaccharide could be the reason for its neuroprotective effect [91].

In a recent study, Kumar et al. found that in DJ-1βΔ93 flies, a model of PD in *Drosophila melanogaster*, dietary supplementation of Spirulina leads to a significant decrease in SOD and CAT activities, improving lifespan and locomotor activity. Supplementation of Spirulina and its active component C-PC individually and independently reduced cellular stress marked by the deregulation of heat shock protein 70 expression and Jun-N-terminal kinase signaling [127].

### 3.2. Spirulina in AD

AD is the most common age-related neurodegenerative disease. It has been estimated that by 2050, more than 100 million individuals around the world will become ill with this disease [128].

AD is a disease that exhibits several possible causal mechanisms, including oxidative stress, chronic inflammation, impaired insulin metabolism and cholinergic deficiency [129].

Pathognomonic signs of AD include the presence of senile plaques with Aβ peptides in the extracellular environment, chronic neuroinflammation and neurofibrillary tangles comprising tau proteins [130].

It has been hypothesized that soluble forms of Aβ oligomers are responsible for the neurodegeneration of brain regions, independent of amyloid plaque deposits [131] and, furthermore, it has been observed that oligomeric Aβ peptides are more destructive than Aβ monomers and fibrillar forms in the brain [132,133]. Consequently, compounds that can inhibit the accumulation of Aβ deposits in the brain are considered potential therapeutic agents [134] and, in recent decades, many researchers have engaged in the development of natural bioactive substances such as Spirulina to cure neurodegenerative diseases such as AD [135]. Scientific evidence has shown the positive effects of Spirulina in AD (Figure 2). In a recent work, Luo and Jing demonstrated that C-PC is able to hinder the amyloid formation process of bovine serum albumin (BSA), inhibiting conformational conversion (α-helices and β-sheets) of BSA [136].

Koh et al. studied the effects of SM70EE in mice with cognitive impairment induced by an intracerebroventricular injection of Aβ1-42. Researchers found that SM70EE reduced Aβ1-42 levels in the hippocampus and inhibited factors associated with APP processing. Additionally, SM70EE suppressed acetylcholinesterase activity, increased the expression levels of GPx and GR, and consequently, the glutathione level. Furthermore, it prevented glycogen synthase kinase-3β (GSK-3β) phosphorylation. SM770EE increased BDNF and potentiated the BDNF/PI3K/Akt signaling pathway, which induced the suppression of GSK-3β phosphorylation, consequently downregulating β-site APP-cleaving enzyme 1 (BACE1), which in turn inhibited the processing of APP [92].

In a transgenic *Caenorhabditis elegans* worm model of AD, Singh et al. showed that phycocyanin could perform its neuroprotective action in AD by blocking the activity of β-secretase [137], which—with γ-secretase—generates Aβ by cleavage of APP [138]. In AD, the increase in soluble Aβ oligomeric species, aggregated Aβ plaques and neurofibrillary tangles of hyperphosphorylated Tau proteins leads to activation of microglia and thus to neuroinflammation [79,139,140]. Several studies supported the protective role of C-PC through the reduction in inflammation and neuronal apoptosis in response to neurodegeneration caused by oligomeric Aβ in mice [141]. Pham et al. demonstrated that dose- and time-dependent *S. platensis* extract inhibits histone deacetylase (HDAC) protein levels and shows anti-inflammatory activity [142]. In another article, it was demonstrated that in the hippocampus of mice treated with oligomeric Aβ peptides, the administration of C-PC increased miRNA-335 levels with reduced apoptosis and attenuated the expression of HDAC3 [141], which represents a hopeful therapeutic target for AD treatment, because inhibition of HDAC3 in the hippocampal tissue reduced amyloid plaque load and Aβ levels and attenuated microglia activation in the brains of APP/PS1 mice [143]. Furthermore, according to Zhu et al. [143], in the hippocampus of 6- and 9-month-old APP/PS1 mice, HDAC3 does not affect the levels of phosphorylated Tau protein. In contrast, Janczura et al. argue that RGFP-966, a selective HDAC3 silencing agent, reduces not only the Aβ protein load but also hyperphosphorylated Tau [144], suggesting an effect of C-PC on Tau protein. In fact, another study on male albino Wistar rats, treated with nicotine for two months, showed that *Spirulina platensis*-lipopolysaccharides are able to prevent phosphorylation of Tau protein [97].

Experimental evidence confirmed that biomarkers of endothelial dysfunction, such as intercellular adhesion molecule 1 (ICAM-1), are increased in AD; endothelial cells, astrocytes and microglia show enhanced levels of ICAM-1 in pathological states, and this plays a role in neuritic plaques on neurite growth and neurodegeneration [145,146,147]. Moreover, chemokines like CXCL2 work via the chemokine receptor (CXCR2) in neuronal cells, where they are involved in the pathophysiology of AD [148]. In ischemic rats treated with PCB, the mRNA expression levels of both markers were strongly reduced [93], suggesting that a similar counteracting effect on AD may improve ICAM-1 and CXCL2-mediated damage in this disease.

Agrawal et al. studied the role of phycocyanin in improving cognitive dysfunction in rats subjected to intracerebroventricular induction of Streptozotocin (STZ). In this experimental model of AD, phycocyanin treatment significantly attenuated neuroinflammation and enhanced BDNF as well as insulin-like growth factor-1 (IGF-1) levels that increased regulation of insulin receptor substrate-1 (IRS-1). Binding of insulin (INS) to its receptor (IR) contributes to autophosphorylation and activation of the IR tyrosis kinase that phosphorylates IRS-1. IRS-1 binds primarily to phosphatidylinositol 3-kinase (PI3K), and phosphorylated PI3K leads to activation of 3-phosphoinositide-dependent protein kinase-1 (PDK1), which in turn phosphorylates serine/threonine protein kinase B (AKT). Phycocyanin has been shown to increase INS-IR binding affinity, IRS-1 activation of PI3K and AKT, resulting in initiation of the IRS/PI3K/AKT pathway, and slightly reduced expression of the phosphatase and tensin homolog (PTEN) gene, an inhibitor of the PI3K/AKT pathway [94].

### 3.3. Spirulina in MS

MS is a chronic inflammatory and neurodegenerative disease of CNS with a possible autoimmune etiology [149]. Neurological deficits in MS patients result from two pathogenic events: acute inflammatory demyelination and axonal injury [150,151].

Currently, drugs used for MS therapeutic treatment aim to suppress the immune system and therefore it is necessary to develop new drugs that act more specifically on the neurodegenerative mechanisms of the disease, and assure the integrity and functioning of myelin, neurons and glial cells [152]. The effects of Spirulina in MS treatment have been the subject of several studies that demonstrated its potential in the treatment of this disease (Figure 2). In particular, phycocyanin seems to be able to protect axons against demyelination. Rats treated with C-PC showed compressed, solid and squashed myelin and no signs of axonal breakdown, characteristics comparable to control animals [153].

Interesting, MS relapses are thought to be due to activated self-reactive myelin-specific T lymphocytes present in the CNS and recruitment of peripheral mononuclear phagocytes, producing inflammation, brain and spinal cord edema, and demyelination [154].

The most used experimental model for the study of this disease is the Experimental Autoimmune Encephalomyelitis (EAE), capable of reproducing the main neuroimmunological and histopathological aspects of MS [155,156]. Rodent EAE studies suggest that acute immunological episodes are auto-limited by Treg, a heterogeneous T cell family implicated in maintaining auto-antigen tolerance [157]. Pentón-Rol et al. demonstrated that C-PC is able to trigger the induction of the Treg subset, as evaluated also in PBMC from MS patients [153]. In EAE mice, C-PC was able to ameliorate the clinical deterioration of animals due to the significant decrease in inflammatory infiltrates in spinal cord tissue, with a reduction in Mac-3 positive activated macrophages/microglia and CD3-positive T cells in the lesions. Moreover, C-PC up-regulated the expression of a set of genes (Mal, Mog and Mobp, Nkx6-2, Nkx2-2, Bmpa and the transcription factor, Olig1) related to remyelination, gliogenesis and axon-glia processes, and significantly reduced the expression of genes involved in the process of demyelination, including CD44 and PPAR. C-PC was also able to reduce both the levels of MDA and PP and the CAT/SOD ratio [158]. More recently, in rat and mice with EAE, oral C-PC administration not only improved EAE clinical progression but also re-established motor function and motor coordination. PCB administration resulted in a significant reduction in the expression levels of pro-inflammatory cytokines like IL-6 and IFN-γ in the brain, positively modulated oxidative stress markers, and preserved the integrity of myelin sheaths in the brain [95]. C-PC is a selective COX-2 inhibitor [22], and inhibition of this enzyme has been confirmed to defend oligodendrocyte precursor cells under various damage conditions [159,160]. Ultimately, therapies that exhibit antioxidant and anti-inflammatory effects, induce Treg and protect axons against demyelination, as demonstrated for C-PC treatment, could represent hope for MS.

## 4. Conclusions

The decrement in normal antioxidant and anti-inflammatory defense systems, which occurs in various conditions, makes the brain more vulnerable to the damaging effects of ROS or RNS and plays an important role in most neurological disorders (AD, PD, MS, lesions inflammatory and senility).

In recent decades, researchers have focused on the development of new drugs, in particular natural bioactive products, which are able to counteract redox imbalance and consequent cell damage without causing neuronal toxicity, for the treatment of neurological and neurodegenerative diseases. Spirulina microalgae represent a rich source of antioxidant and anti-inflammatory substances that is able not only to support the development of the nervous system and physiologic brain functions, compensating for nutritional deficiencies, but also promote a beneficial immune response, reducing the harmful consequences of an overactive immune system.

This review summarizes the latest findings on the neuroprotective role of Spirulina, its positive effects on glial cell activation, and on treatment of neurodegenerative diseases, in particular PD, AD and MS. Several lines of evidence testify to peculiar neuroprotection mechanisms, including antioxidant and anti-inflammatory activities in the brain parenchyma, which make Spirulina a potential pharmacological agent in the prevention and treatment of these neurological disorders. However, despite the numerous and encouraging scientific evidence both in vitro and in vivo, additional studies are needed to clarify the mechanisms of action of Spirulina.

## Figures and Tables

**Figure 1 nutrients-14-00676-f001:**
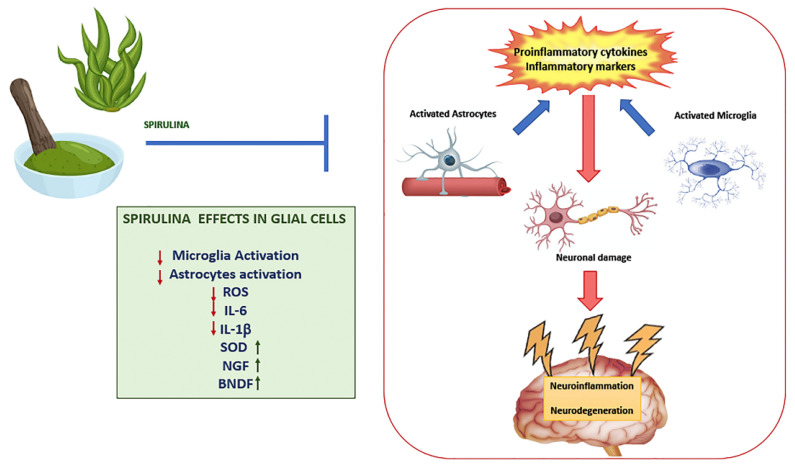
Role of Spirulina in glial cells. Spirulina exerts anti-inflammatory activity with consequent neuronal damage and the onset of neuroinflammatory and/or neurodegenerative disorders, reducing (↓) glial cell activation and proinflammatory molecules, and up-regulating (↑) superoxide dismutase (SOD) and neurotrophic factors such as brain-derived neurotrophic factor (BDNF) and nerve growth factor (NGF).

**Figure 2 nutrients-14-00676-f002:**
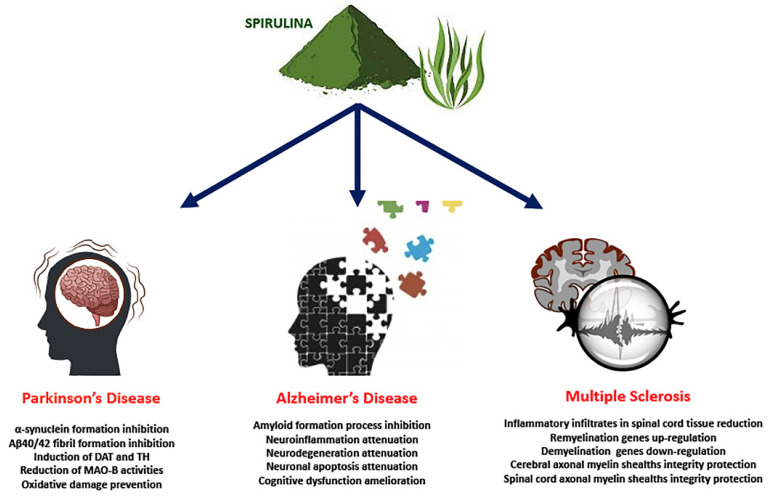
Effects of Spirulina in neurodegenerative diseases.

**Table 1 nutrients-14-00676-t001:** Summary of in-vivo experimental evidence suggesting that Spirulina or its components cross the BBB.

Component/Route of Administration	Animal Model	Summary of Results	Ref.
Spirulina/Orally	Adult male Sprague–Dawley rats with cerebral ischemia	Reduction of infarction area in the cerebral cortex	[48]
		Increase in locomotor activity	
		Decline in TUNEL positive cells and caspase-3 activity	
Spirulina/Orally	Protein malnourished Sprague Dawley female rats	Increased cerebral cortical thickness	[64]
		Reduction in oxidative brain damage	
		Reduction in astrocyte and microglia activation	
PCB/Intraperitoneally	Male Wistar rats with cerebral ischemia	Reduction in the area of cerebral infarction	[65]
		Normalized expression of myelin basic protein and enzyme CNPase	
		Preserved vitality of cerebral cortex neurons	
Spirulina maxima 70% ethanol extract (SM70EE)/Orally	Male ICR mice treated with scopolamine	Reduction in learning and memory deficits	[67]
C-PC/Intraperitoneally	Male Wistar rats treated with tributyltin chloride (TBTC)	Increased C-PC bioavailability in cerebral cortical tissue	[83]
		Reduction in astrocyte and microglia activation	
		Reduction in oxidative stress and inflammation	
C-PC/Orally	Male Sprague Dawley rats treated with kainic acid	Reduction in microglia and astroglia activation	[85]
		Reduced incidence of neurobehavioral changes	
Spirulina/Orally	Male SW mice treated with kainic acid	Reduction of neuron damage in CA3 hippocampal region	[86]
Spirulina/Orally	Male Fisher 344 rats treated with α-Syn in substantia nigra	Reduction in the number of activated microglial cell	[88]
		Increased expression of the fractalkine receptor (CX3CR1) on microglia	
Spirulina/Orally	Male Wistar rats treated with 6-OHDA	Reduction in oxidative stress	[89]
		Preserved dopamine levels in the striatum	
		Normalized locomotor activity	
Spirulina/Orally	male C-57 black mice treated with MPTP	Partial reduction of dopamine content in the striatum	[90]
		Reduction in oxidative stress	
Polysaccharide from Spirulina/Intraperitoneally	C57BL/6J mice treated with MPTP	Increased mRNA expression of dopamine transporter and tyrosine hydroxylase	[91]
		Increased SOD and GPx activity	
Spirulina maxima 70% ethanol extract (SM70EE)/Orally	Male ICR mice treated with Aβ1–42	Reduced oxidative stress	[92]
		Increased GSH, GPx1 and GR levels in the hippocampus	
PCB/Intraperitoneally	Male Wistar rats subjected to permanent bilateral occlusion of the common carotid arteries	Modulated 190 genes associated to immunological and inflammatory processes	[93]
Phycocyanin/Intraperitoneally	Female Wistar rats treated with Streptozotocin	Reduction in neuroinflammation	[94]
		Improved levels of BDNF, IGF-1, BCL-2 and ChAT	
		Improved gene expression of IRS-1, PI3-K, AKT	
C-PC/Orally	Lewis rats with EAE	Restoration of motor function	[95]
		Reduced oxidative damage	
		Preserved the integrity of myelin sheaths	
PCB/Orally	C57BL/6 mice with EAE		[95]
		Improved clinical status of animals	
		Reduction in IL-6 and IFN-γ expression in the brain	
C-PC/Orally or Intraperitoneally	Male Mongolian gerbils with global cerebral ischemia	Reduction in infarct volume	[96]
		Decreased neuronal damage	
		Reduction in malondialdehyde (MDA), peroxidation potential (PP) and FRAP levels	
S. platensis-LPS (S.LPS)/Intraperitoneally	Male albino Wistar rats treated with nicotine	Enhancement in antioxidant enzymes’ activities	[97]
		Improved level of TNF-α, IL-17 and NF-κB in brain tissues	
		Prevention of Tau protein phosphorylation	

## Data Availability

Not applicable.

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
