# Peer review of "Beneficial Effects of Spirulina Consumption on Brain Health"

_nutrients, 2022, doi:10.3390/nu14030676_

Round 1
Reviewer 1 Report
The paper is generally well written and structured. However, in my opinion, the authors should address some comments.
My main concern is about how Spirulina exhibited some beneficial effects. For example, please explain how Spirulina protects the activity of the cellular antioxidant enzymes total GPx, GPx-Se, and GR and how increasing the reduced glutathione. The same for the other studies described in the manuscript. In other words, the authors need to explain the specific mechanisms of action of Spirulina or their components
Please increase the resolution of the Figures used in the manuscript.
Author Response
ANSWERS TO REVIEWER 1
The paper is generally well written and structured. However, in my opinion, the authors should address some comments.
My main concern is about how Spirulina exhibited some beneficial effects. For example, please explain how Spirulina protects the activity of the cellular antioxidant enzymes total GPx, GPx-Se, and GR and how increasing the reduced glutathione. The same for the other studies described in the manuscript. In other words, the authors need to explain the specific mechanisms of action of Spirulina or their components
Thank you for the critical comment. The authors added the mechanisms of action of Spirulina. In detail:
At line 126, instead of:
“In SH-SY5Y neuroblastoma cells, Bermejo-Bescós et al. observed that S. platensis protean extract (rich in phycobiliproteins) and phycocyanin exert the antioxidant activity by protecting the activity of the cellular antioxidant enzymes total GPx, GPx-Se and GR and by increasing reduced glutathione in cells against oxidative stress induced by iron, suggesting that S. platensis protein extract may interfere with radical-mediated cell death and be useful in those diseases in which iron is implicated in neuropathology [57].”
The new sentence is:
Bermejo-Bescós et al. studied, in a neuroblastoma cell line (SH-SY5Y), the action of the S. platensis protein extract on oxidative stress caused by iron, characterized by the exhaustion of the enzyme activities of the GSH pathway, reduction of GSH and increase of the glutathione disulfide (GSSG)/GSH ratio. GSH is an antioxidant enzyme which, by the action of the enzyme glutathione peroxidase (GPx), reduces H2O2 to H2O and oxidizes to GSSG, which in turn is reduced to GSH by the enzyme glutathione reductase (GR) [56]. Fe2+ can lead to the production of hydroxyl or alkoxy radicals, which both the S. platensis protein extract and phycocyanin have been shown to be able to eliminate, preserving the activity of antioxidant enzymes. Furthermore, Spirulina has been shown to be an iron chelator, preventing the reaction with oxygen or peroxides. This study showed that S. platensis protein extract protects the activity of total GPx, GPx-Se and GR and increases reduced GSH, thanks to scavenger activity of free radicals and to iron chelator ability, thus increasing cell viability [57].
At line 180, instead of
“Another study showed that 70% ethanol extract of S. maxima (SM70EE) prevents trimethyltin neurotoxicity, such as oxidative stress, neurotransmitter dysfunction and neuronal death through promoting activation of brain-derived neurotrophic factor (BDNF)/cyclic AMP-responsive element-binding protein (CREB) neuroprotective signaling pathways in HT-22 neuronal cells [67]”
The new sentence is:
“Another study showed that 70% ethanol extract of S. maxima (SM70EE) prevents trimethyltin (TMT) neurotoxicity, such as oxidative stress, neurotransmitter dysfunction and neuronal death. TMT induces the release of brain-derived neurotrophic factor (BDNF), which phosphorylates tropomyosin-related kinase receptor type B (TrkB) resulting in expression of the cyclic AMP responsive element binding protein (CREB) and increase in BDNF, an important factor for neuronal survival. Furthermore, TMT increases acetylcholinesterase (AChE) which in turn reduces the expression of BDNF. The neurotoxicity that is triggered stimulates PARP cleavage, which is a protein associated with apoptosis and therefore the death of neuronal cells. SM70EE is able to inhibit AChE activity and PARP cleavage and to increase the expression of p-CREB, p-TrkB, and BDNF, promoting the activation of BDNF/CREB neuroprotective signaling pathways in HT-22 neuronal cells [67].”
At line 393, instead of
“Koh et al. studied the effects of SM70EE in mice with cognitive impairment induced by an intracerebroventricular injection of Aβ1-42. Researchers found that SM70EE reduced Aβ1-42 levels in the hippocampus and inhibited factors associated with amyloid precursor protein processing. Additionally, SM70EE suppressed acetylcholinesterase activity, increased glutathione levels, promoted activation of the BDNF/phosphatidylinositol-3 kinase/serine/threonine protein kinase signaling pathway, and also inhibited the phosphorylation of glycogen synthase kinase-3β [131].”
The new sentence is:
“Koh et al. studied the effects of SM70EE in mice with cognitive impairment induced by an intracerebroventricular injection of Aβ1-42. Researchers found that SM70EE reduced Aβ1-42 levels in the hippocampus and inhibited factors associated with APP processing. Additionally, SM70EE suppressed acetylcholinesterase activity, increased expression lev-els of GPx and GR and consequently glutathione level and, furthermore, it prevented gly-cogen synthase kinase-3β (GSK-3β) phosphorylation. SM770EE increased BDNF and po-tentiated the BDNF/PI3K/Akt signaling pathway, which induces the suppression of GSK-3β phosphorylation, consequently downregulation of β-site APP-cleaving enzyme 1 (BACE1) which in turn inhibits the processing of APP [92].”
At line 442, instead of
“In this experimental model of AD, phycocyanin treatment significantly attenuated neuroinflammation and enhanced BDNF as well as IGF-1 levels that potently induced the up regulation of IRS-1 to further escalate PI3K/AKT phosphorylation, resulting in initiation of IRS/PI3K/AKT pathway [144].”
The new sentence is:
“In this experimental model of AD, phycocyanin treatment significantly attenuated neuroinflammation and enhanced BDNF as well as insulin-like growth factor-1 (IGF-1) levels that increased regulation of insulin receptor substrate-1 (IRS-1). Binding of insulin (INS) to its receptor (IR) contributes to autophosphorylation and activation of the IR tyrosis kinase that phosphorylates IRS-1. IRS-1 binds primarily to phosphatidylinositol 3-kinase (PI3K) and phosphorylated PI3K leads to activation of 3-phosphoinositide-dependent protein kinase-1 (PDK1), which in turn phosphorylates serine/threonine protein kinase B (AKT). Phycocyanin has been shown to in-crease INS-IR binding affinity, IRS-1 activation of PI3K and AKT, resulting in initiation of IRS/PI3K/AKT pathway, and slightly reduced expression of phosphatase and tensin homolog (PTEN) gene, an inhibitor of the PI3K/AKT pathway [94].
Please increase the resolution of the Figures used in the manuscript.
Thank you for your suggestion. Authors increased Figure resolution.
Reviewer 2 Report
In this interesting, the authors summarized recent advances in the beneficial effects of Spirulina on the health of central nervous system. I have several concerns, that should they be addressed would produce a more compelling review.
- Can you discuss how the protein content of Spirulina cross the brain blood barrier (BBB) with more details in the introduction part? It would be nice to list all the evidence suggesting that the active component can cross BBB since it is crucial for orally administrated nutrients to take effect in the brain.
- The authors documented the role of Spirulina on glia cells activation, I wonder is there any evidence suggesting that Spirulina can work on monocytes and monocyte-derived macrophages during common brain disorders like AD and PD? and what is the role of Spirulina on the BBB damage?
- Since Spirulina has beneficial effects on several neuronal disorders and it can activate glia cells, what is the current knowledge about how Spirulina activate glia cells and convert them into the neuro protective status. Do the active microglia and astrocytes slow down the pathology by increasing phagocytosis ability?
- The authors discussed the role of Spirulina on A-beta in depth, however, what is the role of Spirulina on Tau? Can it influence the aggregation and other pathological modifications (eg. hyper-phosphorylation) of Tau protein?
Author Response
ANSWER TO REVIEWER 2
In this interesting, the authors summarized recent advances in the beneficial effects of Spirulina on the health of central nervous system. I have several concerns, that should they be addressed would produce a more compelling review.
- Can you discuss how the protein content of Spirulina cross the brain blood barrier (BBB) with more details in the introduction part? It would be nice to list all the evidence suggesting that the active component can cross BBB since it is crucial for orally administrated nutrients to take effect in the brain.
Thank you for the comment. In accordance with this suggestion, we inserted the requested information as reported in Table 1. The new sentence is reported at line 259 of the manuscript:
“The possibility that the active components of Spirulina are able to cross the BBB and perform their function in the brain is suggested by in vivo experimental evidence (Table 1). Only Mitra et al. evaluated the bioavailability of C-PC in homogenates of cerebral cortical tissue, by measuring absorbance at 620 nm, in Wistar rats that received intraperitoneal injections of C-PC [83].”
- The authors documented the role of Spirulina on glia cells activation, I wonder is there any evidence suggesting that Spirulina can work on monocytes and monocyte-derived macrophages during common brain disorders like AD and PD? and what is the role of Spirulina on the BBB damage?
Thank you for your comment. In the literature the authors have not found articles suggesting a role of Spirulina on monocytes and monocyte-derived macrophages during neurodegenerative disorders; except one study by Pentón-Rol et al on a mouse model MOG35-55 of EAE, which demonstrated that intraperitoneal administration of C-PC induced a reduction in Mac-3 activated macrophages/microglia levels in white matter lesions [158], as already reported in the section “3.3. Spirulina in MS”.
Regarding the role of Spirulina on the BBB damage, the authors added:
At line 245 “The protective role of C-PC on the permeability of the BBB was evaluated in terms of BBB interruption induced by TBTC. C-PC failed to fully restore BBB disruption, although it was able to restore p-glycoprotein (PGP) and claudin-5 levels and partially enzyme activity of tissue remodeling such as calpain and matrix metalloproteinase [83]. The protective role of C-PC on damaged BBB is also suggested by a small vessel disease (SVD) study, in which McCarty reported that oxidative stress generated by NADPH oxidase may contribute to BBB damage and PCB from Spirulina may improve the course of SVD by suppressing the activity of NADPH oxidase [84].”
- Since Spirulina has beneficial effects on several neuronal disorders and it can activate glia cells, what is the current knowledge about how Spirulina activate glia cells and convert them into the neuro protective status. Do the active microglia and astrocytes slow down the pathology by increasing phagocytosis ability?
Thank you for the critical comment. In the literature, there is little scientific evidence on the action of Spirulina on the phagocytosis capacity of glial cells. However, the authors inserted information about this action. At line 156, it was reported: ”In particular, Spirulina prevented PMN-induced astrogliosis by effectively reducing the astrocytic population labeled with glial fibrillary acidic protein (GFAP) and S100β and with enlarged cell body and thick processes, a morphology that characterizes activated astrocytes. In addition, it was reported that Spirulina caused a reduction of CR3 and MHC II expression on microglia cells that lose the typical shape of activated phagocytic cells.“
At line 233, instead of: “In male Wistar rats treated with tributyltin chloride, an environmental pollutant and potent biocide, C-PC effectively reduced ROS generation, decreased astroglia activation and significantly restored microglial activity apart from its considerable protective effect on redox signaling, stress proteins and inflammatory molecules [84].”
The new sentence is: “In male Wistar rats treated with tributyltin chloride (TBTC), an environmental pollutant and potent biocide, C-PC effectively reduced ROS generation, decreased astroglia activation, counteracting the morphological alteration, and upregulation of GFAP, indices of astroglial activation. In addition, it significantly restored microglial activity, adjusting the expression of the cellular marker CD11b and the formation of LC3-II and beclin (markers of autophagy), reduced by treatment with TBTC, apart from its considerable protective effect on redox signaling, stress proteins and inflammatory molecules [83].”
At line 254, authors added: “evaluated by studying, in the hippocampus, the expression of the heat shock protein 27kD on astrocytes and of the peripheral benzodiazepine receptor on microglia,” and at line 282, “evaluated by studying the expression of GFAP”
- The authors discussed the role of Spirulina on A-beta in depth, however, what is the role of Spirulina on Tau? Can it influence the aggregation and other pathological modifications (eg. hyper-phosphorylation) of Tau protein?
Thank you for the critical comment. At line 425, authors added: “Furthermore, according to Zhu et al [143138] in the hippocampus of 6- and 9-month-old APP/PS1 mouse, HDAC3 does not affect the levels of phosphorylated Tau protein. In contrast, Janczura et al argue that RGFP-966, a selective HDAC3 silencing agent, reduces not only the Aβ protein load but also hyperphosphorylated Tau [144], suggesting a role of C-PC on Tau protein. In fact, another study on male albino Wistar rats, treated with nicotine for two months, showed that Spirulina platensis-lipopolysaccharides is able to prevent phosphorylation of Tau protein [97].”

Round 2
Reviewer 1 Report
The manuscript has improved substantially and most of the reviewers' questions and concerns have been addressed